# Stochastic Dominance and Omega Ratio: Measures to Examine Market Efficiency, Arbitrage Opportunity, and Anomaly

**Xu Guo [1], Xuejun Jiang [2] and Wing-Keung Wong [3,4,5,6,\*]**

[1] School of Statistics, Beijing Normal University, Beijing 100875, China; xguo12@bnu.edu.cn
[2] Department of Mathematics, South University of Science and Technology of China, Shenzhen 518055, China; jiangxj@sustc.edu.cn
[3] Department of Finance and Big Data Research Center, Asia University, Taichung 41354, Taiwan
[4] Department of Economics and Finance, Hang Seng Management College, Hong Kong, China
[5] Department of Economics, Lingnan University, Hong Kong, China
[6] Department of Finance, College of Management, Asia University, 500, Lioufeng Rd., Wufeng, Taichung 41354, Taiwan
[\*] Correspondence: wong@asia.edu.tw

**Abstract:** Both stochastic dominance and Omegaratio can be used to examine whether the market is efficient, whether there is any arbitrage opportunity in the market and whether there is any anomaly in the market. In this paper, we first study the relationship between stochastic dominance and the Omega ratio. We find that second-order stochastic dominance (SD) and/or second-order risk-seeking SD (RSD) alone for any two prospects is not sufficient to imply Omega ratio dominance insofar that the Omega ratio of one asset is always greater than that of the other one. We extend the theory of risk measures by proving that the preference of second-order SD implies the preference of the corresponding Omega ratios only when the return threshold is less than the mean of the higher return asset. On the other hand, the preference of the second-order RSD implies the preference of the corresponding Omega ratios only when the return threshold is larger than the mean of the smaller return asset. Nonetheless, first-order SD does imply Omega ratio dominance. Thereafter, we apply the theory developed in this paper to examine the relationship between property size and property investment in the Hong Kong real estate market. We conclude that the Hong Kong real estate market is not efficient and there are expected arbitrage opportunities and anomalies in the Hong Kong real estate market. Our findings are useful for investors and policy makers in real estate.

**Keywords:** stochastic dominance; Omega ratio; risk averters; risk seekers; utility maximization; market efficiency; anomaly

## 1. Introduction

It is well known that the standard deviation is not a good measure of risk because it penalizes upside deviation, as well as downside deviation. Additionally, it is also poor at measuring risk with asymmetric payoff profiles. The poor performance of the standard deviation will lead to poor performance of the Sharpe ratio, which establishes a relationship between the ratio of return versus volatility (Kapsos et al. (2014); Guastaroba et al. (2016)). A number of studies developed some theories that propose to circumvent the limitations. For example, Homm and Pigorsch (2012) develop an economic performance measure based on Aumann and Serrano's (2008) index of

riskiness. They prove that the proposed economic performance measure is consistent with first- and second-order stochastic dominance (SD). Keating and Shadwick (2002) propose to use the Omega ratio, the probability weighted ratio of gains versus losses to a prospect or the ratio of upside returns (good) relative to downside returns (bad), to replace the Sharpe ratio to measure the risk return performance of a prospect. Thus, the Omega ratio considers all moments, while the Sharpe ratio considers only the first two moments of the return distribution in the construction. According to Caporin et al. (2016), Bellini et al. (2017) and the references provided therein, the Omega ratios are strongly related to expectiles, which are a type of inverse of the Omega ratio and present interesting properties as risk measures. Guastaroba et al. (2016) discuss the advantages of using the Omega ratio further. Thus, the Omega ratio has been commonly used by academics and practitioners as noted by Kapsos et al. (2014) and the references therein.

It is well known that the SD theory can be used to examine whether the market is efficient, whether there is any arbitrage opportunity in the market, and whether there is any anomaly in the market (Sriboonchitta et al. (2009); Levy (2015)), and thus, academics are interested in checking whether there is any relationship between any risk measure with SD. The work from Darsinos and Satchell (2004) and others can be used to establish the relationship between the second-order SD (SSD) and the Omega ratio. By using two counterexamples, we first demonstrate that SSD and/or second-order risk-seeking SD (SRSD) alone for any two prospects is not sufficient to imply Omega ratio dominance (OD) and that the Omega ratio of one asset is always greater than that of the other one. We then extend the work of Darsinos and Satchell (2004) and others by proving that the preference of SSD (for risk averters) implies the preference of the corresponding Omega ratios are selected only when the return threshold is less than the mean of the higher return asset. On the other hand, the preference of SRSD (for risk seekers) implies the preference of the corresponding Omega ratios only when the return threshold is larger than the mean of the smaller return asset. Lastly, we develop the relationship between the first-order SD (FSD) and the Omega ratio in such a way that the preference of FSD for any investor with increasing utility functions does imply the preference of the corresponding Omega ratios for any return threshold.

Qiao and Wong (2015) apply SD tests to examine the relationship between property size and property investment in the Hong Kong real estate market. They do not find any FSD relationship in their study. Tsang et al. (2016) extend their work to reexamine the relationship between property size and property investment in the same market. They suggest to analyze both rental and total yields and find the FSD relationship of rental yield in adjacent pairings of different housing classes in Hong Kong. Based on their analysis on both rental and total yields, they conclude that investing in a smaller house is better than a bigger house. We note that analyzing both rental and total yields is not sufficient to draw such a conclusion. To circumvent the limitation, we extend their work by applying the Omega ratio to examine the relationship between property size and property investment in the Hong Kong real estate market. In addition to analyzing the rental yield, we recommend analyzing the price yields of different houses. We find that a smaller house dominates a bigger house in terms of rental yield, and there is no dominance between smaller and bigger houses in price yield. Our findings lead us to conclude that regardless of whether the buyers are risk averse or risk seeking, they will not only achieve higher expected utility, but also obtain higher expected wealth when buying smaller properties. This implies that the Hong Kong real estate market is not efficient, and there are expected arbitrage opportunities and anomalies in the Hong Kong real estate market. Our findings are useful for real estate investors in their investment decision making and useful to policy makers in real estate for their policy making to make the real estate market become efficient.

The rest of this paper is organized as follows: Section 2 presents the formal definitions of the SD rules and Omega ratios. We then show our main results about the consistency of Omega ratios with respect to the SD in Section 3. In Section 4, we discuss how to apply the theory developed in this paper to examine whether the market is efficient, whether there is any arbitrage opportunity in the market

and whether there is any anomaly in the market. An illustration of the Hong Kong housing market is included in Section 5. The final section offers our conclusion.

## 2. Definitions of Stochastic Dominance and Omega Ratios

We first define cumulative distribution functions (CDFs) for $X$ and $Y$:

$$F_Z^{(1)}(\eta) = F_Z(\eta) = P(Z \leq \eta) \text{, for } Z = X, Y . \tag{1}$$

We define the second-order integral of $Z$, $F_Z^{(2)}$,

$$F_Z^{(2)}(\eta) = \int_{-\infty}^{\eta} F_Z^{(1)}(\xi)d\xi \text{ for } Z = X, Y ; \tag{2}$$

and define the second-order reverse integral, $F_Z^{(2)R}$, of $Z$

$$F_Z^{(2)R}(\eta) = \int_{\eta}^{\infty} (1 - F_Z^{(1)}(\xi))d\xi \text{ for } Z = X, Y . \tag{3}$$

If $Z$ is the return, then $F_Z^{(1)}(\eta)$ is the CDF of the return up to $\eta$ and $F_Z^{(2)}(\eta)$ is the second-order integral of $Z$ up to $\eta$, that is the probability of the CDF of the return up to $\eta$, and $F_Z^{(2)R}(\eta)$ is the second-order reverse integral of $Z$ up to $\eta$, that is the reverse integration of the reverse CDF of the return up to $\eta$. We call $F_Z^{(i)}$ the $i$-th-order integral of $Z$, which will be used to define the SD theory for risk averters (see, for example, Quirk and Saposnik (1962)). On the other hand, we call $F_Z^{(i)R}$ the $i$-th-order reversed integral, which will be used to define the SD theory for risk seekers (see, for example, Hammond (1974)). Risk averters typically have a preference for assets with a lower probability of loss, while risk seekers have a preference for assets with a higher probability of gain. When choosing between two assets $X$ or $Y$, risk averters will compare their corresponding $i$-th order SD integrals $F_X^{(i)}$ and $F_Y^{(i)R}$ and choose $X$ if $F_X^{(i)}$ is smaller, since it has a lower probability of loss. On the other hand, risk seekers will compare their corresponding $i$-th order RSD integrals $F_X^{(i)R}$ and $F_Y^{(i)R}$ and choose $X$ if $F_X^{(i)R}$ is larger since it has a higher probability of gain.

Following the definition of stochastic dominance (Hanoch and Levy (1969)), prospect $X$ first-order stochastically dominates prospect $Y$:

$$\text{if and only if } F_X^{(1)}(\eta) \leq F_Y^{(1)}(\eta) \text{ for any } \eta \in R, \tag{4}$$

which is denoted by $X \succeq_{FSD} Y$; prospect $X$ second-order stochastically dominates prospect $Y$:

$$\text{if and only if } F_X^{(2)}(\eta) \leq F_Y^{(2)}(\eta) \text{ for any } \eta \in R, \tag{5}$$

which is denoted by $X \succeq_{SSD} Y$. Here, FSD and SSD denote first- and second-order stochastic dominance, respectively.

Next, we follow Levy (2015) to define risk-seeking stochastic dominance (RSD)[1] for risk seekers. Prospect $X$ stochastically dominates prospect $Y$ in the sense of second-order risk seeking:

$$\text{if and only if } F_X^{(2)R}(\eta) \geq F_Y^{(2)R}(\eta) \text{ for any } \eta \in R, \tag{6}$$

which is denoted by $X \succeq_{SRSD} Y$. Here, SRSD denotes second-order RSD.

---

[1]　Levy (2015) denotes it as RSSD, while we denote it as RSD.

Quirk and Saposnik (1962), Hanoch and Levy (1969), Levy (2015) and Guo and Wong (2016) have studied various properties of stochastic dominance (for risk averters), while Hammond (1974), Meyer (1977), Stoyan and Daley (1983), Li and Wong (1999), Wong and Li (1999), Wong (2007), Levy (2015) and Guo and Wong (2016) have developed additional properties of risk-seeking stochastic dominance for risk seekers. One important property for SD is that SSD and SRSD are equivalent to the expected-utility maximization for (second-order) risk-averse and risk-seeking investors, respectively, while FSD is equivalent to the expected-utility/wealth maximization for any investor with increasing utility functions.

We turn to define $\Omega_X(\eta)$ as follows:

$$\Omega_X(\eta) = \frac{\int_\eta^\infty (1 - F_X(\xi))d\xi}{\int_{-\infty}^\eta F_X(\xi)d\xi}. \tag{7}$$

Here, $\eta$ is called the return threshold. For any investor, returns below (above) her/his return threshold are considered as losses (gains). Thus, the Omega ratio is the probability weighted ratio of gains to losses relative to a return threshold.

According to Darsinos and Satchell (2004), we can also rewrite $\Omega_X(\eta)$ as follows:

$$\Omega_X(\eta) = \frac{F_X^{(2)R}(\eta)}{F_X^{(2)}(\eta)} = \frac{F_X^{(2)}(\eta) - (\eta - \mu_X)}{F_X^{(2)}(\eta)} = 1 + \frac{\mu_X - \eta}{F_X^{(2)}(\eta)}. \tag{8}$$

We state the following Omega ratio dominance (OD) rule by using the Omega ratio:

**Definition 1.** *For any two prospects X and Y with Omega ratios, $\Omega_X(\eta)$ and $\Omega_Y(\eta)$, respectively, X is said to dominate Y by the Omega ratio or X is said to Omega ratio dominate Y, denote by:*

$$X \succeq_{OD} Y \quad \text{if} \quad \Omega_X(\eta) \geq \Omega_Y(\eta) \text{ for any } \eta \in R. \tag{9}$$

## 3. Consistency Results

We will use the term "theorem" to state new results obtained in this paper and "proposition" to state some well-known results. Some academics may believe that the SSD is consistent with the Omega ratio because they assert the following:

$$\text{if} \quad X \succeq_{SSD} Y, \quad \text{then} \quad \Omega_X(\eta) \geq \Omega_Y(\eta) \quad \text{for any} \quad \eta \in R, \tag{10}$$

where $\Omega_X(\eta)$ is the Omega ratio for $X$ defined in (7) or (8). The above assertion is in Darsinos and Satchell (2004) and others. We first establish the following property to state that the argument in (10) may not be correct:

**Property 1.** *SSD alone is not sufficient to imply $\Omega_X(\eta) \geq \Omega_Y(\eta)$ for any $\eta$.*

Property 1 implies that the assertion made by Darsinos and Satchell (2004) and others may not be always correct. We construct the following example to support the argument stated in Property 1.

**Example 1.** *Consider two prospects X and Y having the following distributions:*

$$X = 10 \quad \text{with prob. } 1, \quad \text{and} \quad Y = \begin{cases} 1 & \text{with prob. } 2/3 \\ 11 & \text{with prob. } 1/3 \end{cases}. \tag{11}$$

*Then, we get $\mu_X = 10$ and $\mu_Y = 13/3$ and obtain the following:*

$$F_X^{(2)}(\eta) = \begin{cases} 0 & \text{if } \eta < 10 \\ \eta - 10 & \text{if } \eta \geq 10 \end{cases}, \quad F_Y^{(2)}(\eta) = \begin{cases} 0 & \text{if } \eta < 1 \\ 2(\eta - 1)/3 & \text{if } 1 \leq \eta < 11 \\ \eta - 13/3 & \text{if } \eta \geq 11 \end{cases},$$

$$F_X^{(2)R}(\eta) = \begin{cases} 10 - \eta & \text{if } \eta < 10 \\ 0 & \text{if } \eta \geq 10 \end{cases}, \quad F_Y^{(2)R}(\eta) = \begin{cases} 13/3 - \eta & \text{if } \eta < 1 \\ (11 - \eta)/3 & \text{if } 1 \leq \eta < 11 \\ 0 & \text{if } \eta \geq 11 \end{cases}.$$

*It follows that $F_X^{(2)}(\eta) \leq F_Y^{(2)}(\eta)$, for all $\eta \in R$. That is, $X \succeq_{SSD} Y$. However, for any $10 \leq \eta < 11$, we have $F_X^{(2)R}(\eta) \equiv 0 < F_Y^{(2)R}(\eta)$. Recalling the definition of $\Omega_X(\eta)$, we can conclude that $\Omega_X(\eta) \equiv 0 < \Omega_Y(\eta)$ for any $10 \leq \eta < 11$, and thus, the statement "$\Omega_X(\eta) \geq \Omega_Y(\eta)$ for any $\eta$" does not hold.*

To complement Property 1, we establish the following property:

**Property 2.** *SRSD alone is not sufficient to imply $\Omega_X(\eta) \geq \Omega_Y(\eta)$ for any $\eta$.*

We construct the following example to support the argument stated in Property 2.

**Example 2.** *Consider two prospects X and Y as follows:*

$$X = \begin{cases} 2 & \text{with prob. } 1/2 \\ 8 & \text{with prob. } 1/2 \end{cases} \quad \text{and} \quad Y = \begin{cases} 3 & \text{with prob. } 2/3 \\ 6 & \text{with prob. } 1/3 \end{cases}. \tag{12}$$

*We have $\mu_X = 5$ and $\mu_Y = 4$ and obtain the following:*

$$F_X^{(2)}(\eta) = \begin{cases} 0 & \text{if } \eta < 2 \\ (\eta - 2)/2 & \text{if } 2 \leq \eta < 8, \\ \eta - 5 & \text{if } \eta \geq 8 \end{cases} \quad F_Y^{(2)}(\eta) = \begin{cases} 0 & \text{if } \eta < 3 \\ 2(\eta - 3)/3 & \text{if } 3 \leq \eta < 6, \\ \eta - 4 & \text{if } \eta \geq 6 \end{cases}$$

$$F_X^{(2)R}(\eta) = \begin{cases} 5 - \eta & \text{if } \eta < 2 \\ 4 - \eta/2 & \text{if } 2 \leq \eta < 8, \\ 0 & \text{if } \eta \geq 8 \end{cases} \quad F_Y^{(2)R}(\eta) = \begin{cases} 4 - \eta & \text{if } \eta < 3 \\ 2 - \eta/3 & \text{if } 3 \leq \eta < 6. \\ 0 & \text{if } \eta \geq 6 \end{cases}$$

*It follows that $F_X^{(2)R}(\eta) \geq F_Y^{(2)R}(\eta)$, for all $\eta \in R$. This concludes that $X \succeq_{SRSD} Y$. However, for $\eta = 3.3$, we can get:*

$$\Omega_X(\eta) = \frac{F_X^{(2)R}(\eta)}{F_X^{(2)}(\eta)} = \frac{4 - \eta/2}{(\eta - 2)/2} = \frac{8 - \eta}{\eta - 2} = 3.615.$$

$$\Omega_Y(\eta) = \frac{F_Y^{(2)R}(\eta)}{F_Y^{(2)}(\eta)} = \frac{2 - \eta/3}{2(\eta - 3)/3} = \frac{6 - \eta}{2\eta - 6} = 4.5.$$

*That is, $\Omega_X(\eta) < \Omega_Y(\eta)$. In fact, for any $3 < \eta < 7 - \sqrt{13}$, we have $\Omega_X(\eta) < \Omega_Y(\eta)$, and thus, the statement "$\Omega_X(\eta) \geq \Omega_Y(\eta)$ for any $\eta$" does not hold.*

Properties 1 and 2 tell us that SSD and SRSD alone are not sufficient to imply $\Omega_X(\eta) \geq \Omega_Y(\eta)$ for any $\eta$. Then, one may ask: what is the relationship between $\Omega_X(\eta)$ and $\Omega_Y(\eta)$ when there is SSD or SRSD? Guo et al. (2016) and Balder and Schweizer (2017) provide an answer. In this paper, we restate their result to extend the work by Darsinos and Satchell (2004) and others by first deriving the relationship between SSD (for risk averters) and the Omega ratio:

**Proposition 1.** *For any two returns X and Y with means $\mu_X$ and $\mu_Y$ and Omega ratios $\Omega_X(\eta)$ and $\Omega_Y(\eta)$, respectively, if $X \succeq_{SSD} Y$, then $\Omega_X(\eta) \geq \Omega_Y(\eta)$ for any $\eta \leq \mu_X$.*

Now, it is clear that Proposition 1 extends the results of Darsinos and Satchell (2004) by restricting the range of the return threshold. We note that Balder and Schweizer (2017) obtain a similar result of Proposition 1. However, we have independently derived Proposition 1 and reported the results in Guo et al. (2016). Moreover, our proof is different from Balder and Schweizer (2017).

In addition, we also study the relationship of second-order risk-seeking stochastic dominance and the corresponding Omega ratios. A dual result as stated in Theorem 1 is obtained. Finally, the relationship between first-order stochastic dominance and the Omega ratios is established in Corollary 2. Some simple examples (Examples 1 and 2) are presented to show that SSD or SRSD alone are not sufficient to imply $\Omega_X(\eta) \geq \Omega_Y(\eta)$ for any $\eta$.

Here, we provide a short proof[2] as follows: although it is true that if $X \succeq_{SSD} Y$, then $\mu_X - \eta \geq \mu_Y - \eta$ for any $\eta$. However, the sign of $\mu_X - \eta$ and $\mu_Y - \eta$ can be negative. To be precise, for $\eta > \mu_X \geq \mu_Y$, $0 > \mu_X - \eta \geq \mu_Y - \eta$. In this situation, we can get:

$$\frac{\mu_X - \eta}{F_X^{(2)}(\eta)} \leq \frac{\mu_X - \eta}{F_Y^{(2)}(\eta)}.$$

Furthermore, we note that:

$$\frac{\mu_Y - \eta}{F_Y^{(2)}(\eta)} = \frac{\mu_Y - \mu_X}{F_Y^{(2)}(\eta)} + \frac{\mu_X - \eta}{F_Y^{(2)}(\eta)} \leq \frac{\mu_X - \eta}{F_Y^{(2)}(\eta)}.$$

Consequently, we cannot determine the sign of $\frac{\mu_X - \eta}{F_X^{(2)}(\eta)} - \frac{\mu_Y - \eta}{F_Y^{(2)}(\eta)}$. Thus, we cannot determine the sign of $\Omega_X(\eta) - \Omega_Y(\eta)$. However, for any $\eta \leq \mu_X$, we can have $\mu_X - \eta \geq 0$, and thus, we have:

$$\frac{\mu_X - \eta}{F_X^{(2)}(\eta)} \geq \frac{\mu_X - \eta}{F_Y^{(2)}(\eta)} \geq \frac{\mu_Y - \eta}{F_Y^{(2)}(\eta)}.$$

This implies that $\Omega_X(\eta) \geq \Omega_Y(\eta)$, and thus, the assertion of Proposition 1 holds.    □

In the proof of Proposition 1, one could conclude that if $X \succeq_{SSD} Y$, then $\Omega_X(\eta) \geq \Omega_Y(\eta)$ for any $\eta \leq \mu_X$. However, for $\eta > \mu_X$, we cannot determine which one is larger if we are using SSD. However, one could consider employing the SD (RSD) theory for risk seeking (refer to Equation (6)) in the study. By doing so, we establish the following theorem to state the relationship between the SRSD and Omega ratio:

**Theorem 1.** *For any two returns X and Y with means $\mu_X$ and $\mu_Y$ and Omega ratios $\Omega_X(\eta)$ and $\Omega_Y(\eta)$, respectively, if $X \succeq_{SRSD} Y$, then $\Omega_X(\eta) \geq \Omega_Y(\eta)$ for any $\eta \geq \mu_Y$.*

Here, we give a short proof as follows: assume that $X \succeq_{SRSD} Y$. This is equivalent to $F_X^{(2)R}(\eta) = \int_\eta^\infty (1 - F_X(\xi))d\xi \geq F_Y^{(2)R}(\eta)$. Recall that $F_X^{(2)R}(\eta) = F_X^{(2)}(\eta) - (\eta - \mu_X) \geq 0$. This yields the following equation:

$$\frac{1}{\Omega_X(\eta)} = \frac{F_X^{(2)}(\eta)}{F_X^{(2)R}(\eta)} = \frac{F_X^{(2)R}(\eta) + (\eta - \mu_X)}{F_X^{(2)R}(\eta)} = 1 + \frac{\eta - \mu_X}{F_X^{(2)R}(\eta)}.$$

---

2    We note that our proof is different from that of Balder and Schweizer (2017).

Further, we note that $X \succeq_{SRSD} Y$ implies $\mu_X \geq \mu_Y$. Thus, for $\eta \geq \mu_Y$, we obtain:

$$\frac{\eta - \mu_X}{F_X^{(2)R}(\eta)} = \frac{\eta - \mu_Y}{F_X^{(2)R}(\eta)} + \frac{\mu_Y - \mu_X}{F_X^{(2)R}(\eta)} \leq \frac{\eta - \mu_Y}{F_X^{(2)R}(\eta)} \leq \frac{\eta - \mu_Y}{F_Y^{(2)R}(\eta)}.$$

In other words, we can get $\Omega_X(\eta) \geq \Omega_Y(\eta)$ for any $\eta \geq \mu_Y$, and thus, the assertion of Theorem 1 holds.

□

We note that Darsinos and Satchell (2004) assert that SSD is consistent with the Omega ratio; that is, the relationship in Equation (10) holds. However, we find that the consistency of SSD and the Omega ratio holds only when we restrict the range of return threshold, as stated in our Proposition 1 and Theorem 1. From Proposition 1 and Theorem 1, one could then derive the following theorem to state the relationship between the FSD and Omega ratio:

**Theorem 2.** *If the SSD and SRSD hold, then the Omega ratio dominance also holds. In particular, this is the case when the FSD holds.*

We give a short proof as follows: if $X \succeq_{FSD} Y$, by using the hierarchy property (Levy (1992, 1998, 2015); Sriboonchitta et al. (2009)), we obtain both $X \succeq_{SSD} Y$ and $X \succeq_{SRSD} Y$. From Proposition 1 and Theorem 1, we have $\Omega_X(\eta) \geq \Omega_Y(\eta)$ for any $\eta \leq \mu_X$ and $\eta \geq \mu_Y$. Since $\mu_X \geq \mu_Y$, we have $\Omega_X(\eta) \geq \Omega_Y(\eta)$ for any $\eta \in R$, and thus, the assertion of Theorem 2 holds. □

## 4. Testing Market Efficiency, Arbitrage Opportunity and Anomaly

In this section, we will discuss how to apply the theory developed in this paper to examine whether the market is efficient, whether there is any arbitrage opportunity in the market and whether there is any anomaly in the market. To do so, we consider the following four pairs of hypotheses:

$$H_0^{SSD}: \ X \nsucc_{SSD} Y \quad \text{versus} \quad H_1^{SSD}: X \succ_{SSD} Y \tag{13}$$

$$H_0^{SRSD}: X \nsucc_{SRSD} Y \quad \text{versus} \quad H_1^{SRSD}: X \succ_{SRSD} Y \tag{14}$$

$$H_0^{FSD}: X \nsucc_{FSD} Y \quad \text{versus} \quad H_1^{FSD}: X \succ_{FSD} Y \tag{15}$$

$$H_0^{OD}: X \nsucc_{OD} Y \quad \text{versus} \quad H_1^{OD}: X \succ_{OD} Y \tag{16}$$

To test whether there is any SSD in two assets as stated in (13), we can apply Proposition 1 to test whether $\Omega_X(\eta) \geq \Omega_Y(\eta)$ for any $\eta \leq \mu_X$. If this is true, then we could have $X \succ_{SSD} Y$. Similarly, to test whether there is any SRSD in two assets as stated in (14), we can apply Theorem 1 to test whether $\Omega_X(\eta) \geq \Omega_Y(\eta)$ for any $\eta \geq \mu_Y$. If this is true, then we could have $X \succ_{SRSD} Y$. Last, to test whether there is any FSD in two assets as stated in (15), we can apply Theorem 2 and Definition 1 to test whether $X \succeq_{OD} Y$. If this is true, then we could have $X \succ_{FSD} Y$. Readers may ask: why should we test $H_1^{SSD}$ in (13), $H_1^{SRSD}$ in (14), $H_1^{FSD}$ in (15), and $H_1^{OD}$ in (16)? The answer is that we want to test whether there is any arbitrage opportunity in the market, whether there is any anomaly and whether the market is efficient. We first discuss testing arbitrage opportunity and anomaly, and, thereafter, discuss testing market efficiency and investor rationality in the next subsections.

### 4.1. Arbitrage Opportunity and Anomaly

It is well known from the market efficiency hypothesis that if one can get an abnormal return, then the market is considered inefficient, and there could exist arbitrage opportunity and anomaly. Thus, in order to test arbitrage opportunity and anomaly, one can apply Theorem 2 and Definition 1 to test $H_1^{OD}$ in (16) and check whether $X \succeq_{OD} Y$. If $X \succeq_{OD} Y$, then applying Theorem 2, we can conclude that $X \succ_{FSD} Y$ could be true. Jarrow (1986) and Falk and Levy (1989) have claimed that if FSD exists, under certain conditions, arbitrage opportunities also exist, and investors will increase not only their

expected utilities, but also their wealth if they shift from holding the dominated asset to the dominant one. One may consider it a financial anomaly.

However, Wong et al. (2008) have shown that if FSD exists statistically, arbitrage opportunities may not exist, but investors can increase their expected utilities, as well as their expected wealth, but not their wealth if they shift from holding the dominated asset to the dominant one. In this paper, we call this situation "expected arbitrage opportunity" or "arbitrage opportunity in expectation"; this means that if $X \succeq_{OD} Y$ appears many times and if investors could buy $X$ and short sell $Y$ each time, then on average, they could not only increase their expected utility, but also increase their expected wealth. In this situation, one may believe that there could be arbitrage opportunity and anomaly.

Falk and Levy (1989), Bernard and Seyhun (1997) and Larsen and Resnick (1999) comment that if there exists first-order dominance of a particular asset over another, but the dominance does not last for a long period, market efficiency and market rationality cannot be rejected. In general, the first-order dominance should not last for a long period of time because if the market is rational and efficient, then market forces will adjust the market so that there is no FSD. For example, if Property A dominates Property B at the FSD, then all investors would buy Property A and sell Property B. This will continue driving up the price of Property A relative to Property B, until the market price of Property A relative to Property B is high enough to make the marginal investor indifferent between Properties A and B. In this situation, we conclude that the market is still efficient and that investors are still rational. In the traditional theory of market efficiency, if one is able to earn an abnormal return for a considerable length of time, the market is considered inefficient. If new information is either quickly made public or anticipated, the opportunity to use the new information to earn an abnormal return is of very limited value. On the other hand, if the first-order dominance can hold for a long time and all investors can increase their expected wealth by switching their asset choice, we claim that the market is inefficient and that investors are irrational. However, sometimes FSD could still be held for a long period if investors do not realize such dominance exists or there are some reasons for the investors to buy the dominated asset. For example, investors could prefer to buy a bigger property for their status, even if the price is too high. If the FSD relationship among some assets still exists over a long period of time, then we could have arbitrage opportunity and anomaly, that market is inefficient and that investors are not rational.

*4.2. Market Efficiency and Rationality*

In last section, if $H_1^{OD}$ in (16) such that $X \succeq_{OD} Y$ is not rejected over a long period of time, then we conclude that there could be arbitrage opportunity and anomaly, that the market is inefficient, and that investors are not rational. Nonetheless, if $H_1^{OD}$ in (16) is rejected, should we conclude that the market is efficient and that investors are rational? Here, we would like to recommend academics and practitioners to further examine the higher order SD, say, for example, the second-order SD, before they conclude that the market is efficient.

Falk and Levy (1989) have argued that, given two assets, X and Y, if by switching from X to Y (or by selling X short and holding Y long), an investor can increase expected utility, the market is inefficient. SSD does not imply any arbitrage opportunity, but it does imply the preference of one asset over another by risk-averse investors. For example, if we apply Proposition 1 to test whether $\Omega_A(\eta) \geq \Omega_B(\eta)$ for any $\eta \leq \mu_A$ and find that it is true, then we could have $A \succ_{SSD} B$, and thus, Property A dominates Property B by SSD. In this situation, one would not make an expected profit by switching from Property B to Property A, but switching would allow risk-averse investors to increase their expected utility. In this situation, could we conclude that the property market is not efficient?

We suggest that this claim could be made if one believes that the market only contains risk-averse investors. However, it is well known that the market could have other types of investors (see, for example, Friedman and Savage (1948), Markowitz (1952), Thaler and Johnson (1990), Broll et al. (2010) and Egozcue et al. (2011) for more discussion). Under the assumption that the market could contain more than one type of investor, such as risk averters, as well as risk seekers,

in this situations, academics could apply Theorem 1 to test whether $\Omega_B(\eta) \geq \Omega_A(\eta)$ for any $\eta \geq \mu_A$. If this is true, then we could have $B \succ_{SRSD} A$, and thus, Property A dominates Property B by SSD and Property B dominates Property A by SRSD. If this is the case, then risk averters could prefer to buy Property A, while risk seekers prefer to invest in Property B. Then, equilibrium could be reached in the sense that both Properties A and B can be sold well, and there is no upward or downward pressure on the price of both Properties A and B, while both risk averters and risk seekers could get what they want. Under these conditions, Qiao et al. (2012) argue that the market is still efficient and investors are still rational. On the other hand, if Property A dominates Property B by both SSD and SRSD, then one could conclude that the market is inefficient. However, if Property A dominates Property B by both SSD and SRSD, then Property A dominates Property B by FSD. We have discussed this case in the above.

## 5. Illustration

Investment in property is important in both consumption and investment decisions (Henderson and Ioannides (1987)). Ziering and McIntosh (2000) argue that housing size is important in determining the risk and return of housing and conclude that the largest class of housing provides investors with the highest return and the greatest volatility. However, Flavin and Nakagawa (2008) document that investing in larger houses does not reduce risk, while Kallberg et al. (1996) show that smaller property offers impactful diversification benefits for investment portfolios with high return aspirations. On the other hand, Cannon et al. (2006) explain housing returns by volatility, price level and stock-market risk, and Ghent and Owyang (2010) investigate supply and demand to explain movements in the housing market.

The housing market in Hong Kong plays a very important role in the Hong Kong economy (Haila (2000)), and Hong Kong is one of the most expensive housing markets in the world in terms of both prices and rents (Tsang et al. (2016)). Qiao and Wong (2015) apply SD tests to examine the relationship between property size and property investment in the Hong Kong real estate market. They do not find any FSD relationship in their study. Tsang et al. (2016) extend their work to reexamine the relationship between property size and property investment in the same market and find the FSD relationship in rental yield in any adjacent pairing of the five well-defined housing classes in Hong Kong. In empirical studies, very few studies could discover the existence of any FSD relationship, and it is very important to obtain the FSD relationship (if there is any) because this information is very helpful to investors. For example, the findings from Tsang et al. (2016) imply that by shifting investing from the largest class of housing to the smallest class of housing, investors could obtain higher expected utility, as well as higher expected wealth from rental income.

In this paper, we extend their work by applying the Omega ratio to examine the relationship between property size and property investment in the Hong Kong real estate market. We recommend that analysts apply the Omega ratio to examine whether there is any FSD relationship between any pair of variables being studied because it is easier to obtain the Omega ratio. The Omega ratio could serve as a complementary tool for the FSD test, and thus, we recommend that analysts use both the Omega ratio and FSD test in their analysis. The existence of dominance from both the Omega ratio and FSD test could assert the existence of the FSD relationship between the variables being examined. In addition, our illustration could also serve our purpose to demonstrate whether the theory developed in this paper holds true.

In order to readdress the issue studied by Tsang et al. (2016), we first use the same rental yield data used in Tsang et al. (2016) to compare monthly property-market rental yields in private domestic units of five different housing classes from January 1999–December 2013 in Hong Kong. The data are obtained from the Rating and Valuation Department of the Hong Kong SAR. The monthly rental yields for each class are calculated by dividing the average rent within the class by the average sale price for houses in the class for that month. Private domestic units are defined as independent dwellings with separate cooking facilities and bathrooms (and/or lavatories). They are sub-divided into five classes

by reference to floor area: Class A salable area less than 40 m$^2$; Class B salable area of 40–69.9 m$^2$; Class C salable area of 70–99.9 m$^2$; Class D salable area of 100–159.9 m$^2$; and Class E salable area of 160 m$^2$ or above.

To analyze the rental yield and to illustrate Theorem 2, we set *A* = rental yield of Class A and *E* = rental yield of Class E and present the summary statistics of the rental yields for Classes *A* and *E* in Table 1.

**Table 1.** Summary statistics for *X* and *Y*.

| Class | Mean | std | Skewness | Kurtosis | JBtest | *t*-test/F-test |
|---|---|---|---|---|---|---|
| *A* | 0.0041 | 0.0008 | −0.1957 | 1.9615 | 1 | 0.0000 |
| *E* | 0.0028 | 0.0008 | 0.4521 | 1.8709 | 1 | 0.7059 |

Note: *A* = the rental yield of Class A, *E* = the rental yield of Class E, and std = standard deviation. *t*- and F-tests report the *p*-values of the tests.

We first test the following hypotheses:

$$H_0^\mu: \ \mu_A = \mu_E \quad \text{versus} \quad H_1^\mu: \mu_A > \mu_E \tag{17}$$

for rental yield. The result of the *t*-test in Table 1 concludes that the mean rental yield of *A* is significantly higher than that of *E*. Thereafter, we test the following hypotheses:

$$H_0^\sigma: \ \sigma_A = \sigma_E \quad \text{versus} \quad H_1^\sigma: \sigma_A < \sigma_E \tag{18}$$

for rental yield. The result of the *F*-test in Table 1 does not reject that the variances of the rental yields of both *A* and *E* are the same. Applying the mean-variance rule for risk averters Markowitz (1952) that *A* is better than *E* if $\mu_A \geq \mu_E$, $\sigma_A \leq \sigma_E$ and there is at least one strictly inequality, we conclude that risk averters prefer Property *A* to Property *E* based on rental yield. On the other hand, if we apply the mean-variance rule for risk seekers (Wong (2006, 2007); Guo et al. (2017)) that *A* is better than *E* provided that $\mu_A \geq \mu_E$, $\sigma_A \geq \sigma_E$ and there is at least one strict inequality, we conclude that risk seekers prefer Property *A* to Property *E* based on rental yield under the condition that *A* and *E* belong to the same location-scale family or the same linear combination of location-scale families Wong (2006, 2007). Nonetheless, this conclusion cannot imply the existence of the first-order SD relationship between Properties *A* and *E* based on rental yield if *A* and *E* do not belong to the same location-scale family or the same linear combination of location-scale families. To circumvent the limitation, this paper recommends that academics and practitioners use the Omega ratio rule as discussed in this paper. Thus, we turn to applying the Omega ratio rule to analyze whether there is any first-order SD relationship between Properties *A* and *E* based on rental yield.

We note that for the existence of the Omega ratio, we need $Z < \eta$ with $Z = A, E$. To satisfy this condition, we choose $\eta > \max(\min(A), \min(E))$. In addition, the term $(\eta - Z)_+$ should not be too small. If not, the Omega ratios will be very large. Thus, in this illustration, we set $\eta \geq \max(\min(A), \min(E)) + 0.5\%$. Furthermore, for $\eta \geq \max(\max(A), \max(E))$, we have $(Z - \eta)_+ \equiv 0$. Thus, we set the upper-bound for $\eta$ as $\max(\max(A), \max(E))$. We exhibit the plot in Figure 1.

From the figure, it is clear that $\Omega_A(\eta) \geq \Omega_E(\eta)$ for any $\eta \in R$. We skip displaying plots of other pairs of variables because all the plots draw the same conclusion. We find that Class A dominates Classes B, C, D and E, Class B dominates Classes C, D and E, Class C dominates Classes D and E and Class D dominates Class E, by using the Omega ratio rule. We summarize the results of the Omega ratio dominance in Table 2. The results in the table are read based on row versus column. For example, the cell in Row A and Column B tells us that Class A dominates Class B by the Omega ratio and is denoted by OD, while the cell in Row B and Column A means that Class B does not dominate Class A by the Omega ratio, as denoted by ND.

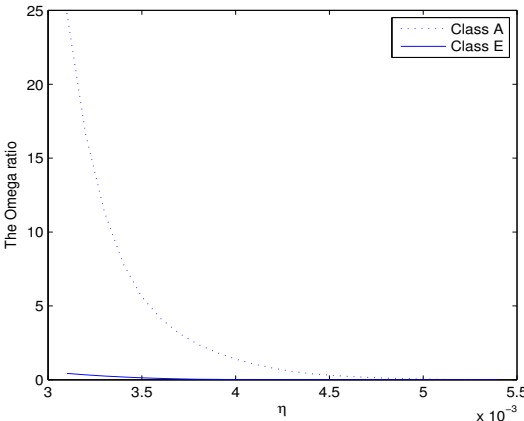

**Figure 1.** The plots of Omega ratios of rental yields of Class A and Class E. The dotted and solid line represent the results of Class A and Class E, respectively.

**Table 2.** Pairwise comparison between rental yields.

| Class | A | B | C | D | E |
|-------|-----|-----|-----|-----|-----|
| A | | OD | OD | OD | OD |
| B | ND | | OD | OD | OD |
| C | ND | ND | | OD | OD |
| D | ND | ND | ND | | OD |
| E | ND | ND | ND | ND | |

OD is Omega ratio dominance defined in Definition 1. ND means no Omega ratio dominance.

To check whether a smaller house (any house in the group with the smaller size) is better than a bigger house (any house in the group with the bigger size), only comparing their rental yields is not good enough. Tsang et al. (2016) suggest analyzing both rental and total yields. Based on their analysis on both rental and total yields, they conclude that investing in a smaller house is better than a bigger house. We note that analyzing both rental and total yields is not sufficient to draw such a conclusion. We explain the reasons as follows: Tsang et al. (2016) find that (a) the smaller house dominates the bigger house in terms of rental yield, and (c) there is no dominance between the smaller and bigger houses in total yield where total yield = rental yield + price yield. Under (a) and (c), it is possible that (b') the bigger house dominates the smaller house in terms of the price yield, and thus, under (a), (b') and (c), we cannot conclude that the smaller house is a better investment than the bigger house. To circumvent the limitation, in addition to analyzing the rental yield, we recommend analyzing the price yield as follows: We set $A$ = price yield of Class A and $E$ = price yield of Class E and present the summary statistics of the price yields for Classes $A$ and $E$ in Table 3.

**Table 3.** Summary statistics of the price yield for Classes $A$ and $E$.

| Class | Mean | std | Skewness | Kurtosis | JB test | *t*-test/F-test |
|-------|--------|--------|----------|----------|---------|-----------------|
| $A$ | 0.0054 | 0.0233 | −0.2194 | 3.6554 | 0 | 0.5327 |
| $E$ | 0.0056 | 0.0308 | −0.1495 | 3.9346 | 1 | 0.0001 |

Note: $A$ = the price yield of Class A and $E$ = the price yield of Class E. *t*- and F-tests report the *p*-values of the tests.

We first test the null hypothesis $H_0^{\mu}$ that $\mu_A = \mu_E$ versus the alternative hypothesis $H_1^{\mu}$ that $\mu_A > \mu_E$ as shown in (17) for the price yield. The result of the *t*-test in Table 3 does not reject that the mean price yields for *A* and *E* are the same. Thereafter, we test the null hypothesis $H_0^{\sigma}$ that $\sigma_A = \sigma_E$ versus the alternative hypothesis $H_1^{\sigma}$ that $\sigma_A < \sigma_E$ as shown in (18) for the price yield. The result of the *F*-test in Table 3 concludes that the variance of the price yield of *A* is significantly smaller that that of *E*. Thus, applying the mean-variance rules, we can conclude that risk averters prefer to invest in *A* rather than *E*, but risk seekers are indifferent between *A* and *E*. Nonetheless, this conclusion cannot imply any first-order SD relationship between Properties *A* and *E* based on the price yield. In this paper, we recommend that academics and practitioners use the Omega ratio rule as discussed in this paper.

Continuing with our analysis in the rental yield, we find that when $\eta \geq 0.0054$, $\Omega_A(\eta)$ is smaller than $\Omega_E(\eta)$, while when $\eta < 0.0054$, $\Omega_A(\eta)$ is larger. Thus, there is no OD relationship between *A* and *E*. To illustrate our results empirically, we set $\eta \in [0.0054, \max(\max(A), \max(E))]$. The related results are exhibited in Figure 2. From this figure, it is clear that the Omega ratio of Class E is larger than that of Class A. For the Omega ratio dominance, different from the analysis for the rent yields, there is no dominance relationship between A and E in terms of the price yield by using the Omega ratio, and thus, we conclude that there is no FSD relationship between A and E in terms of the price yield.

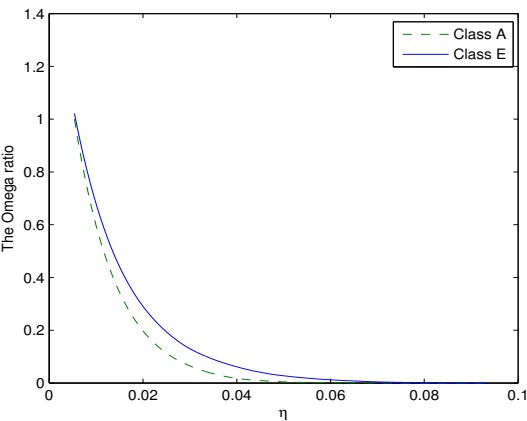

**Figure 2.** The plots of Omega ratios of the price yield of Class A and Class E. The dotted and solid line represent the results of Class A and Class E, respectively.

Tsang et al. (2016) find that Class A SSD dominates Class E in terms of total yield. We conduct the Omega ratio test analysis for this issue. Our findings are consistent with Tsang et al. (2016). Since using both rental yield and price yield could draw the conclusion that investing in the smaller house is better than the bigger house, we skip reporting the OD results for the total yield.

Recall that Tsang et al. (2016) have shown that (a) the smaller house dominates the bigger house in terms of rental yield and (c) there is no dominance between smaller and bigger houses in total yield. Under (a) and (c), it is possible that (b') the bigger house dominates the smaller house in terms of the price yield.

In this paper, we find that (a) the smaller house dominates the bigger house in terms of rental yield, and (b) there is no dominance between smaller and bigger houses in price yield. We note that total yield = rental yield + price yield. Findings (a) and (b) can get either (c) that the smaller house dominates the bigger house in terms of total yield or (c') there is no dominance between smaller and bigger houses in terms of the total yield. No matter under (a), (b) and (c) or under (a), (b) and (c') (actually, we find (c') in our paper), we conclude that regardless of whether the buyers are risk averse or risk seeking, they will not only achieve higher expected utility, but also obtain higher expected wealth when buying smaller properties. This implies that the Hong Kong real estate market is not efficient, and there are expected arbitrage opportunities and anomalies in the Hong Kong real estate

market. Our findings are useful for real estate investors and policy makers in real estate for their policy making to make the real estate market become efficient.

Last, we note that though our paper finds that there exists "expected arbitrage opportunity" in the Hong Kong real estate market, however, it is very difficult, if not impossible, to short sell a property in Hong Kong. Thus, it is not easy to explore this "expected arbitrage opportunity". Nonetheless, if an investor would like to buy a big house to stay in Hong Kong and sell it a couple years later, then, she/he may consider buying a few smaller houses with the same amount of funds in total, rent out all the smaller houses she/he bought, rent a bigger house for her/him to stay and sell all the properties she/he bought as her/his plan a couple of years later. In this way, she/he will get positive net rental income each month (since the rental yield of the smaller house OD dominates that of the bigger house), while the price yield has no difference when she/he sells the big house or the small houses (since there is OD dominance between smaller and bigger houses in terms of price yield). Thus, when she/he sells all her/his properties, she/he still gets net profit by the rental rental if she/he chooses to buy small houses.

## 6. Concluding Remarks

This paper first develops the relationship between the first- and second-order SD with the Omega ratio dominance. We then illustrate the applicability of the theory developed in this paper to examine the relationship between property size and property investment in the Hong Kong real estate market and draw the conclusion that the Hong Kong real estate market is inefficient, and there are expected arbitrage opportunities and anomalies in the Hong Kong real estate market. Our findings are useful for real estate investors and policy makers in real estate for their policy making to make the real estate market become efficient.

We note that the stochastic dominance tests have been well developed by now. For example, one could apply the SD tests developed by Bai et al. (2015) to examine whether there is any FSD, SSD or SRSD relationship between any two prospects. Then, one could apply the theory developed in this paper to draw inference on the preference of the corresponding Omega ratios under different conditions and for different types of investors, including risk averters, risk seekers and investors with increasing utility functions. We note that recently, Hoang et al. (2015) hypothesized that the preference of the Omega ratios implies the preference of the corresponding assets for risk averters or risk seekers. We note that this is not so straight-forward, and this is another good direction of further study in this area (Guo and Wong (2017)). Another direction of related research is to extend Niu et al. (2016, 2017) and others to develop risk measures with a different order of stochastic dominance. This could further be used to examine whether the market is efficient and whether there is arbitrage opportunity in the market.

**Acknowledgments:** The authors are grateful to the Editor and two anonymous referees for constructive comments and suggestions that led to a significant improvement of an early manuscript. The third author would like to thank Robert B. Miller and Howard E. Thompson for their continuous guidance and encouragement. Xu Guo's work is partially supported by the China Postdoctoral Science Foundation (2017M610058), the National Natural Science Foundation of China (No. 11601227 and No. 11626130) and the Natural Science Foundation of Jiangsu Province, China (No. BK20150732). Xuejun Jiang's work is partially supported by the National Natural Science Foundation of China (No. 11101432) and the Natural Science Foundation of Guangdong Province, China (No. 2016A030313856). Wing-Keung Wong's work is partially supported by grants from Asia University, Hang Seng Management College, Lingnan University, Ministry of Science and Technology (MOST), Taiwan, and the Research Grants Council (RGC) of Hong Kong.

**Author Contributions:** Xu Guo presents the basic ideas and obtains the main results in Section 3; Xuejun Jiang conducts the Illustration section; Wing-keung Wong writes the Section 4 and also the whole paper.

**Conflicts of Interest:** The authors declare no conflicts of interests.

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
