# Peer review of "Stochastic Dominance and Omega Ratio: Measures to Examine Market Efficiency, Arbitrage Opportunity, and Anomaly"

_economies, doi:10.3390/economies5040038_

Round 1

Reviewer 1 Report

Report on “Stochastic Dominance and Omega Ratio: Measures to Examine Market Efficiency and Anomaly”

The authors study the relationship between stochastic dominance and Omega ratio. They find that second-order stochastic dominance (SD) and/or second-order risk-seeking SD (RSD) alone for any two prospects is not sufficient to imply Omega-ratio dominance insofar that the Omega ratio of one asset is always greater than that of the other one. They observe that the preference of second-order SD implies the preference of the corresponding Omega ratios only when the return threshold is less than the mean of the higher-return asset. They derive that the preference of the second-order RSD implies the preference of the corresponding Omega ratios only when the return threshold is larger than the mean of the smaller-return asset. They also find that first-order SD does imply Omega-ratio dominance. Thereafter, they examine the relationship between property size and property investment in the Hong Kong real estate market and draw conclusion on market efficiency and anomaly in the Hong Kong real estate market.

The findings in their paper are interesting. I have the following comments to the authors to improve their paper:

1)    The authors claim that both stochastic dominance and Omega ratio can be used to examine whether market is efficient and whether there is any anomaly in the market. However, I find that they have not told readers why both stochastic dominance and Omega ratio can be used to examine whether market is efficient and whether there is any anomaly. They should discuss the issue clearly.

2)    The SD theory (for risk averters) is well known but not the SD theory for risk seekers. The authors should discuss the SD theory for risk seekers more clearly and cite some important work in this area.

3)    It seems that all or part of Proposition 3.1 has been developed partially by Guo, et al. (2016) and Balder and Schweizer (2017). The authors should indicate the difference among Proposition 3.1 and the related results developed by Guo, et al. (2016) and Balder and Schweizer (2017).

4)    The authors claim that they extend the work by Darsinos and Satchell (2004). The authors should state clearly the related work by Darsinos and Satchell (2004) and how their work different from those in Darsinos and Satchell (2004).

5)    Corollary 3.1 is one of the most important findings in the paper and this is new in the literature. Thus, I suggest the authors call it a theorem instead of corollary.

6)    The authors claim that their findings imply that the Hong Kong real estate market is not efficient and there is arbitrage opportunity in the Hong Kong real estate market.

However, the authors have not discussed why SD (and thereafter, the Omega ratio) can be used to test whether the market is efficient and whether there is arbitrage opportunity in the market. The authors should discuss this issue clearly and cite some important literature to support their claim.

7)    The authors apply their theory to examine the relationship between property size and property investment in the Hong Kong real estate market. However, they have not discussed the literature of the relationship between property size and property investment in the Hong Kong real estate market well. I suggest the author discuss the literature clearly and cite some related literature in this area.

8)    The authors should polish their writing to some extent.

Author Response

1)    The authors claim that both stochastic dominance and Omega ratio can be used to examine whether market is efficient and whether there is any anomaly in the market. However, I find that they have not told readers why both stochastic dominance and Omega ratio can be used to examine whether market is efficient and whether there is any anomaly. They should discuss the issue clearly.

 Thank you very much for your advice. We have included in new section (Testing Market Efficiency and Anomaly) to discuss how to apply the theory developed in this paper to examine whether the market is efficient, whether there is any arbitrage opportunity in the market and whether there is any anomaly in the market.

2)    The SD theory (for risk averters) is well known but not the SD theory for risk seekers. The authors should discuss the SD theory for risk seekers more clearly and cite some important work in this area.

 We have discussed the SD theory for risk seekers more clearly and cite some important work in this area in our revised manuscript.

3)    It seems that all or part of Proposition 3.1 has been developed partially by Guo, et al. (2016) and Balder and Schweizer (2017). The authors should indicate the difference among Proposition 3.1 and the related results developed by Guo, et al. (2016) and Balder and Schweizer (2017).

We note that Balder and Schweizer (2017) obtain a similar result of Proposition 3.1 in our paper. However, we have independently derived Proposition 3.1 and reported in Guo et al. (2016). In addition, our proof is different from Balder and Schweizer (2017).

 4)    The authors claim that they extend the work by Darsinos and Satchell (2004). The authors should state clearly the related work by Darsinos and Satchell (2004) and how their work different from those in Darsinos and Satchell (2004).

Darsinos and Satchell (2004) assert that SSD is consistent with the Omegatio ratio; that is, the relationship in Equation (3.1) holds. However, we find that the consistency of SSD and Omega ratio holds only when we restrict the range of return threshold, as stated in Proposition 3.1 and Theorem 3.1 in our paper.

 5)    Corollary 3.1 is one of the most important findings in the paper and this is new in the literature. Thus, I suggest the authors call it a theorem instead of corollary.

 Done

6)    The authors claim that their findings imply that the Hong Kong real estate market is not efficient and there is arbitrage opportunity in the Hong Kong real estate market. However, the authors have not discussed why SD (and thereafter, the Omega ratio) can be used to test whether the market is efficient and whether there is arbitrage opportunity in the market. The authors should discuss this issue clearly and cite some important literature to support their claim.

 Thank you very much for your advice. We have included in new section (Testing Market Efficiency and Anomaly) to discuss how to apply the theory developed in this paper to examine whether the market is efficient, whether there is any arbitrage opportunity in the market, and whether there is any anomaly in the market. We have discussed this issue clearly and cite some important literature to support our claim.

 7)    The authors apply their theory to examine the relationship between property size and property investment in the Hong Kong real estate market. However, they have not discussed the literature of the relationship between property size and property investment in the Hong Kong real estate market well. I suggest the author discuss the literature clearly and cite some related literature in this area.

We have discussed the literature of the relationship between property size and property investment in the Hong Kong real estate market in the revised manuscript.

 8)    The authors should polish their writing to some extent.

We have polished our revised manuscript carefully.

Reviewer 2 Report

I like the general idea of using Omega ratios to examine market properties (and to study theoretically to which extent this is possible). Regarding the presentation, I would suggest to make it a bit more salient what the paper's main contribution is, e.g., by clearly formulating what your proposed test looks like. At present, we directly proceed from Section 3 (Consistency) to Section 4 (Illustration), even though most of the consistency results are known in the literature (as is well acknowledged in the paper). It would help the reader to have a clear main result/message in the paper before the illustrations start.

I have a few concrete suggestions for improving the Illustration section:

- Can you provide a few summary statistics beforehand (such as mean yields per category etc)?

- Could you spell out in more detail how to make an arbitrage in Hong Kong's housing market? I can imagine some stories here - but the classical story from financial markets where agents go long and short in various assets do not apply.

- I would be cautious with statements about agent's utility in this example as utility from housing does not only come from the financial side but also from how well a concrete apartment fits an agents needs.

Minor Points:

- p.3 "increasing utility" not "increased"

- p.3 "Our findings" should be "our findings", same on p. 12.

- p.8 Kappa is usually called a performance measure (not a risk measure). In fact, one could probably delete the three lines about Kappa, as it does not appear in the paper otherwise.

- p.11 Please change the plot style for the two curves in Figure 4.1. These are hard enough to see on the screen and almost impossible to see when printed on paper.

Author Response

1. I would suggest to make it a bit more salient what the paper’s main contribution is, e.g., by clearly formulating what your proposed test looks like.

Thank you very much for your helpful advice. We are more salient on our paper’s main contribution.

2. At present, we directly proceed from Section 3 (Consistency) to Section 4 (Illustration), even though most of the consistency results are known in the literature (as is well acknowledged in the paper). It would help the reader to have a clear main result/message in the paper before the illustrations start.

Thank you very much for your advice. We have included in new section (Testing Market Efficiency and Anomaly) between Consistency and Illustration to tell readers how to use the theory developed in our paper.

3. Can you provide a few summary statistics beforehand (such as mean yields per category etc)?

We have provided two tables of summary statistics for both rental and price yields in Section 5.

4. Could you spell out in more detail how to make an arbitrage in Hong Kong’s housing market? I can imagine some stories here - but the classical story from financial markets where agents go long and short in various assets do not apply.

Thank you very much for your advice. We have stated a way how to make an “arbitrage” in Hong Kong’s housing market in our revised manuscript. However, we would appreciate it very much if you could tell us the stories you imagined to improve our paper.

5. I would be cautious with statements about agent’s utility in this example as utility from housing does not only come from the financial side but also from how well a concrete apartment fits an agents needs.

Thank you very much for your advice. We have discussed this issue in our paper.

6. p.3 ”increasing utility” not ”increased”

Done.

7. p.3 ”Our findings” should be ”our findings”, same on p. 12.

Done.

8. p.8 Kappa is usually called a performance measure (not a risk measure). In fact, one could probably delete the three lines about Kappa, as it does not appear in the paper otherwise.

Thank you very much for your advice. We have deleted the the three lines about Kappa ratio.

9. p.11 Please change the plot style for the two curves in Figure 4.1. These are hard enough to see on the screen and almost impossible to see when printed on paper.

We have made our plots clearer.